# `FAVAS`: Federated AVeraging with ASynchronous clients

## Abstract

In this paper, we propose a novel centralized Asynchronous Federated Learning (FL) framework, `FAVAS` for training Deep Neural Networks (DNNs) in resource-constrained environments. Despite its popularity, "classical" federated learning faces the increasingly difficult task of scaling synchronous communication over large wireless networks. Moreover, clients typically have different computing resources and therefore computing speed, which can lead to a significant bias (in favor of "fast" clients) when the updates are asynchronous. Therefore, practical deployment of FL requires to handle users with strongly varying computing speed in communication/resource constrained setting. We provide convergence guarantees for `FAVAS` in a smooth, non-convex environment and carefully compare the obtained convergence guarantees with existing bounds, when they are available. Experimental results show that the `FAVAS` algorithm outperforms current methods on standard benchmarks.

## 1 Introduction

Federated learning, a promising approach for training models from networked agents, involves the collaborative aggregation of locally computed updates, such as parameters, under centralized orchestration (Konečný et al., 2015; McMahan et al., 2017; Kairouz et al., 2021). The primary motivation behind this approach is to maintain privacy, as local data is never shared between agents and the central server (Zhao et al., 2018; Horváth et al., 2022). However, communication of training information between edge devices and the server is still necessary. The central server aggregates the local models to update the global model, which is then sent back to the devices. Federated learning helps alleviate privacy concerns, and it distributes the computational load among networked agents. However, each agent must have more computational power than is required for inference, leading to a computational power bottleneck. This bottleneck is especially important when federated learning is used in heterogeneous, cross-device applications.

Most approaches to centralized federated learning (FL) rely on synchronous operations, as assumed in many studies (McMahan et al., 2017; Wang et al., 2021). At each global iteration, a copy of the current model is sent from the central server to a selected subset of agents. The agents then update their model parameters using their private data and send the model updates back to the server. The server aggregates these updates to create a new shared model, and this process is repeated until the shared model meets a desired criterion. However, device heterogeneity and communication bottlenecks (such as latency and bandwidth) can cause delays, message loss, and stragglers, and the agents selected in each round must wait for the slowest one before starting the next round of computation. This waiting time can be significant, especially since nodes may have different computation speeds.

To address this challenge, researchers have proposed several approaches that enable asynchronous communication, resulting in improved scalability of distributed/federated learning (Xie et al., 2019;

Chen et al., 2020, 2021; Xu et al., 2021). In this case, the central server and local agents typically operate with inconsistent versions of the shared model, and synchronization in lockstep is not required, even between participants in the same round. As a result, the server can start aggregating client updates as soon as they are available, reducing training time and improving scalability in practice and theory.

**Contributions.** Our work takes a step toward answering this question by introducing FAVAS, a centralized federated learning algorithm designed to accommodate clients with varying computing resources and support asynchronous communication.

- In this paper, we introduce a new algorithm called FAVAS that uses an unbiased aggregation scheme for centralized federated learning with asynchronous communication. Our algorithm does not assume that clients computed the same number of epochs while being contacted, and we give non-asymptotic complexity bounds for FAVAS in the smooth nonconvex setting. We emphasize that the dependence of the bounds on the total number of agents $n$ is improved compared to Zakerinia et al. (2022) and does not depend on a maximum delay.

- Experimental results show that our approach consistently outperforms other asynchronous baselines on the challenging TinyImageNet dataset (Le and Yang, 2015).

Our proposed algorithm FAVAS is designed to allow clients to perform their local steps independently of the server's round structure, using a fully local, possibly outdated version of the model. Upon entering the computation, all clients are given a copy of the global model and perform at most $K \geq 1$ optimization steps based on their local data. The server randomly selects a group of $s$ clients in each server round, which, upon receiving the server's request, submit an *unbiased* version of their progress. Although they may still be in the middle of the local optimization process, they send reweighted contributions so that fast and slow clients contribute equally. The central server then aggregates the models and sends selected clients a copy of the current model. The clients take this received server model as a new starting point for their next local iteration.

## 2 Related Works

Federated Averaging (FedAvg), also known as local SGD, is a widely used approach in federated learning. In this method, each client updates its local model using multiple steps of stochastic gradient descent (SGD) to optimize a local objective function. The local devices then submit their model updates to the central server for aggregation, and the server updates its own model parameters by averaging the client models before sending the updated server parameters to all clients. FedAvg has been shown to achieve high communication efficiency with infrequent synchronization, outperforming distributed large mini-batches SGD (Lin et al., 2019).

However, the use of multiple local epochs in FedAvg can cause each device to converge to the optima of its local objective rather than the global objective, a phenomenon known as client drift. This problem has been discussed in previous work; see (Karimireddy et al., 2020). Most of these studies have focused on synchronous federated learning methods, which have a similar update structure to FedAvg (Wang et al., 2020; Karimireddy et al., 2020; Qu et al., 2021; Makarenko et al., 2022; Mao et al., 2022; Tyurin and Richtárik, 2022). However, synchronous methods can be disadvantageous because they require all clients to wait when one or more clients suffer from high network delays or have more data, and require a longer training time. This results in idle time and wasted computing resources.

Moreover, as the number of nodes in a system increases, it becomes infeasible for the central server to perform synchronous rounds among all participants, and synchrony can degrade the performance of distributed learning. A simple approach to mitigate this problem is node sampling, e.g. Smith et al. (2017); Bonawitz et al. (2019), where the server only communicates with a subset of the nodes in a round. But if the number of stragglers is large, the overall training process still suffers from delays.

Synchronous FL methods are prone to stragglers. One important research direction is based on FedAsync (Xie et al., 2019) and subsequent works. The core idea is to update the global model immediately when the central server receives a local model. However, when staleness is important, performance is similar to FedAvg, so it is suboptimal in practice. ASO-Fed (Chen et al., 2020) proposes to overcome this problem and handles asynchronous FL with local streaming data by

introducing memory-terms on the local client side. AsyncFedED (Wang et al., 2022) also relies on the FedAsync instantaneous update strategy and also proposes to dynamically adjust the learning rate and the number of local epochs to staleness. Only one local updated model is involved in FedAsync-like global model aggregations. As a result, a larger number of training epochs are required and the frequency of communication between the server and the workers increases greatly, resulting in massive bandwidth consumption. From a different perspective, QuAFL (Zakerinia et al., 2022) develops a concurrent algorithm that is closer to the FedAvg strategy. QuAFL incorporates both asynchronous and compressed communication with convergence guarantees. Each client must compute $K$ local steps and can be interrupted by the central server at any time. The client updates its model with the (compressed) central version and its current private model. The central server randomly selects $s$ clients and updates the model with the (compressed) received local progress (since last contact) and the previous central model. QuAFL works with old variants of the model at each step, which slows convergence. However, when time, rather than the number of server rounds, is taken into account, QuAFL can provide a speedup because the asynchronous framework does not suffer from delays caused by stragglers. A concurrent and asynchronous approach aggregates local updates before updating the global model: FedBuff (Nguyen et al., 2022) addresses asynchrony using a buffer on the server side. Clients perform local iterations, and the base station updates the global model only after $Z$ different clients have completed and sent their local updates. The gradients computed on the client side may be stale. The main assumption is that the client computations completed at each step come from a uniform distribution across all clients. Fedbuff is asynchronous, but is also sensitive to stragglers (must wait until $Z$ different clients have done all local updates). Similarly, Koloskova et al. (2022) focus on Asynchronous SGD, and provide guarantees depending on some $\tau_{max}$. Similar to Nguyen et al. (2022) the algorithm is also impacted by stragglers, during the transitional regime at least. A recent work by Fraboni et al. (2023) extend the idea of Koloskova et al. (2022) by allowing multiple clients to contribute in one round. But this scheme also favors fast clients. Liu et al. (2021) does not run on buffers, but develops an Adaptive Asynchronous Federated Learning (AAFL) mechanism to deal with speed differences between local devices. Similar to FedBuff, in Liu et al. (2021)'s method, only a certain fraction of the locally updated models contribute to the global model update. Most convergence guarantees for asynchronous distributed methods depend on staleness or gradient delays (Nguyen et al., 2022; Toghani and Uribe, 2022; Koloskova et al., 2022). Only Mishchenko et al. (2022) analyzes the asynchronous stochastic gradient descent (SGD) independently of the delays in the gradients. However, in the heterogeneous (non-IID) setting, convergence is proved up to an additive term that depends on the dissimilarity limit between the gradients of the local and global objective functions.

## 3 Algorithm

We consider optimization problems in which the components of the objective function (i.e., the data for machine learning problems) are distributed over $n$ clients, i.e.,

$$\min_{w \in \mathbb{R}^d} R(w); \ R(w) = \frac{1}{n} \sum_{i=1}^{n} \mathbb{E}_{(x,y) \sim p_{\text{data}}^i}[\ell(\text{NN}(x, w), y)],$$

where $d$ is the number of parameters (network weights and biases), $n$ is the total number of clients, $\ell$ is the training loss (e.g., cross-entropy or quadratic loss), $\text{NN}(x, w)$ is the DNN prediction function, $p_{\text{data}}^i$ is the training distribution on client $i$. In FL, the distributions $p_{\text{data}}^i$ are allowed to differ between clients (statistical heterogeneity).

Each client maintains three key values in its local memory: the local model $w^i$, a counter $q^i$, and the value of the initial model with which it started the iterations $w_{init}^i$. The counter $q^i$ is incremented for each SGD step the client performs locally until it reaches $K$, at which point the client stops updating its local model and waits for the server request. Upon the request to the client $i$, the local model and counter $q^i$ are reset. If a server request occurs before the $K$ local steps are completed, the client simply pauses its current training process, reweights its gradient based on the number of local epochs (defined by $E_{t+1}^i$), and sends its current *reweighted* model to the server.

In Zakerinia et al. (2022), we identified the client update $w^i = \frac{1}{s+1} w_{t-1} + \frac{s}{s+1} w^i$ as a major shortcoming. When the number of sampled clients $s$ is large enough, $\frac{s}{s+1} w^i$ dominates the update and basically no server term are taken into consideration. This leads to a significant client drift. As a

**Algorithm 1:** FAVAS over $T$ iterations. In red are highlighted the differences with QuAFL.

**Input** : Number of steps $T$, LR $\eta$, Selection Size $s$, Maximum local steps $K$ ;

```
/* At the Central Server                    */
```
**1 Initialize**
**2** | Initialize parameters $w_0$;
**3** | Server sends $w_0$ to all clients;
**4 end**
**5 for** $t = 1, \ldots, T$ **do**
**6** | Generate set $\mathcal{S}_t$ of $s$ clients uniformly at random;
**7** | **for** *all clients* $i \in \mathcal{S}_t$ **do**
**8** | | Server receives $w^i_{unbiased}$ from client $i$;
**9** | **end**
**10** | Update central server model
     $w_t \leftarrow \frac{1}{s+1} w_{t-1} + (\frac{1}{s+1} \sum_{i \in \mathcal{S}_t} w^i_{unbiased})$;
**11** | **for** *all clients* $i \in \mathcal{S}_t$ **do**
**12** | | Server sends $w_t$ to client $i$;
**13** | **end**
**14 end**

```
/* At Client i                              */
```
**15 Initialize**
**16** | Client receives $w_0$ and $K$ from the Server;
**17** | **Local variables** $w^i = w_0, q^i = 0$;
**18 end**
**19 Loop**
**20** | Run ClientLocalTraining() concurrently;
**21** | **When** *Contacted by the Server* **do**
**22** | | Interrupt ClientLocalTraining();
**23** | | Define $\alpha^i$ following (3) ;
**24** | | Send $w^i_{unbiased} := w^i_{init} + \frac{1}{\alpha^i}(w^i - w^i_{init})$ to the server;
**25** | | Receive $w_t$ from the server;
**26** | | Update $w^i_{init} \leftarrow w_t, w^i \leftarrow w_t, q^i \leftarrow 0$;
**27** | | Restart ClientLocalTraining() from zero with updated variables;
**28** | **end**
**29 end**
**30 function** ClientLocalTraining():
**31** | **while** $q^i < K$ **do**
**32** | | Compute local stochastic gradient $\widetilde{g}^i$ at $w^i$;
**33** | | Update local model $w^i \leftarrow w^i - \eta \widetilde{g}^i$;
**34** | | Update local counter $q^i \leftarrow q^i + 1$;
**35** | **end**
**36** | Wait();
**37 end function**

140 consequence, QuAFL does not perform well in the heterogeneous case (see Section 5). Second, one
141 can note that the updates in QuAFL are biased in favor of fast clients. Indeed each client computes
142 gradients at its own pace and can reach different numbers of epochs while being contacted by the
143 central server. It is assumed that clients compute the *same* number of local epochs in the analysis
144 from Zakerinia et al. (2022), but it is not the case in practice. As a consequence, we propose FAVAS to
145 deal with asynchronous updates without favoring fast clients. A first improvement is to update local
146 weight directly with the received central model. Details can be found in Algorithm 1. Another idea
147 to tackle gradient unbiasedness is to reweight the contributions from each of the $s$ selected clients:
148 these can be done either by dividing by the (proper) number of locally computed epochs, or by the
149 expected value of locally computed epochs. In practice, we define the reweight $\alpha^i = \mathbb{E}[E^i_{t+1} \wedge K]$,
150 or $\alpha^i = \mathbf{P}(E^i_{t+1} > 0)(E^i_{t+1} \wedge K)$, where $\wedge$ stands for $\min$. We assume that the server performs
151 a number of training epochs $T \geq 1$. At each time step $t \in \{1, \ldots, T\}$, the server has a model $w_t$.
152 At initialization, the central server transmits identical parameters $w_0$ to all devices. At each time
153 step $t$, the central server selects a subset $\mathcal{S}_t$ of $s$ clients uniformly at random and requests their local
154 models. Then, the requested clients submit their *reweighted* local models back to the server. When
155 all requested models arrive at the server, the server model is updated based on a simple average (see
156 Line 10). Finally, the server multicasts the updated server model to all clients in $\mathcal{S}_t$. In particular, all
157 clients $i \notin \mathcal{S}_t$ continue to run their individual processes without interruption.

158 **Remark 1.** *In* FAVAS*'s setting, we assume that each client* $i \in \{1, ..., n\}$ *keeps a full-precision local*
159 *model* $w^i$. *In order to reduce the computational cost induced by the training process,* FAVAS *can also*
160 *be implemented with a quantization function* $Q$. *First, each client computes backpropagation with*
161 *respect to its quantized weights* $Q(w^i)$. *That is, the stochastic gradients are unbiased estimates of*
162 $\nabla f_i \left( Q\left( w^i \right) \right)$. *Moreover, the activations computed at forward propagation are quantized. Finally,*
163 *the stochastic gradient obtained at backpropagation is quantized before the SGD update. In our*
164 *supplementary experiments, we use the logarithmic unbiased quantization method of Chmiel et al.*
165 *(2021).*

Table 1: How long one has to wait to reach an $\epsilon$ accuracy for non-convex functions. For simplicity, we ignore all constant terms. Each constant $C_\_$ depends on client speeds and represents the unit of time one has to wait in between two consecutive server steps. $L$ is the Lipschitz constant, and $F := (f(w_0) - f_*)$ is the initial conditions term. $a_i, b$ are constants depending on client speeds statistics, and defined in Theorem 3.

| Method | Units of time |
|---|---|
| FedAvg | $\left( \frac{FL\sigma^2 + (1 - \frac{s}{n})KG^2}{sK} \epsilon^{-2} + FL^{\frac{1}{2}}G\epsilon^{-\frac{3}{2}} + LFB^2\epsilon^{-1} \right) C_{FedAvg}$ |
| FedBuff | $\left( FL(\sigma^2 + G^2)\epsilon^{-2} + FL((\frac{\tau_{max}^2}{s^2} + 1)(\sigma^2 + nG^2))^{\frac{1}{2}}\epsilon^{-\frac{3}{2}} + FL\epsilon^{-1} \right) C_{FedBuff}$ |
| AsyncSGD | $\left( FL(3\sigma^2 + 4G^2)\epsilon^{-2} + FLG(s\tau_{avg})^{\frac{1}{2}}\epsilon^{-\frac{3}{2}} + (s\tau_{max}F)^{\frac{1}{2}}\epsilon^{-1} \right) C_{AsyncSGD}$ |
| QuAFL | $\frac{1}{E^2}FLK(\sigma^2 + 2KG^2)\epsilon^{-2} + \frac{n\sqrt{n}}{E\sqrt{Es}}FKL(\sigma^2 + 2KG^2)^{\frac{1}{2}}\epsilon^{-\frac{3}{2}} + \frac{1}{E\sqrt{s}}n\sqrt{n}FBK^2L\epsilon^{-1}$ |
| FAVAS | $FL(\sigma^2 \sum_i^n \frac{a_i}{n} + 8G^2 b)\epsilon^{-2} + \frac{n}{s}FL^2(K^2\sigma^2 + L^2K^2G^2 + s^2\sigma^2 \sum_i^n \frac{a_i}{n} + s^2G^2 b)^{\frac{1}{2}}\epsilon^{-\frac{3}{2}} + nFB^2KLb\epsilon^{-1}$ |

## 4 Analysis

In this section we provide complexity bounds for FAVAS in a smooth nonconvex environment. We introduce an abstraction to model the stochastic optimization process and prove convergence guarantees for FAVAS.

**Preliminaries.** We abstract the optimization process to simplify the analysis. In the proposed algorithm, each client asynchronously computes its own local updates without taking into account the server time step $t$. Here in the analysis, we introduce a different, but statistically equivalent setting. At the beginning of each server timestep $t$, each client maintains a local model $w_{t-1}^i$. We then assume that all $n$ clients *instantaneously* compute local steps from SGD. The update in local step $q$ for a client $i$ is given by:

$$\widetilde{h}_{t,q}^i = \widetilde{g}^i \left( w_{t-1}^i - \sum_{s=1}^{q-1} \eta \widetilde{h}_{t,s}^i \right),$$

where $\widetilde{g}^i$ represents the stochastic gradient that client $i$ computes for the function $f_i$. We also define $n$ independent random variables $E_t^1, \ldots, E_t^n$ in $\mathbb{N}$. Each random variable $E_t^i$ models the number of local steps the client $i$ could take before receiving the server request. We then introduce the following random variable: $\widetilde{h}_t^i = \sum_{q=1}^{E_t^i} \widetilde{h}_{t,q}^i$. Compared to Zakerinia et al. (2022), we do not assume that clients performed the same number of local epochs. Instead, we reweight the sum of the gradients by weights $\alpha^i$, which can be either *stochastic* or *deterministic*:

$$\alpha^i = \begin{cases} \mathbf{P}(E_{t+1}^i > 0)(E_{t+1}^i \wedge K) & \text{\textit{stochastic} version,} \\ \mathbb{E}[E_{t+1}^i \wedge K] & \text{\textit{deterministic} version.} \end{cases} \tag{1}$$

And we can define the *unbiased* gradient estimator: $\check{h}_t^i = \frac{1}{\alpha^i} \sum_{q=1}^{E_t^i \wedge K} \widetilde{h}_{t,q}^i$.

Finally, a subset $\mathcal{S}_t$ of $s$ clients is chosen uniformly at random. This subset corresponds to the clients that send their models to the server at time step $t$. In the current notation, each client $i \in \mathcal{S}_t$ sends the value $w_{t-1}^i - \eta \check{h}_t^i$ to the server. We emphasise that in our abstraction, all clients compute $E_t^i$ local updates. However, only the clients in $\mathcal{S}_t$ send their updates to the server, and each client $i \in \mathcal{S}_t$ sends only the $K$ first updates. As a result, we introduce the following update equations:

$$\begin{cases} w_t = \frac{1}{s+1}w_{t-1} + \frac{1}{s+1}\sum_{i \in \mathcal{S}_t}(w_{t-1}^i - \eta\frac{1}{\alpha^i}\sum_{s=1}^{E_t^i \wedge K} \widetilde{h}_{t,s}^i), \\ w_t^i = w_t, \quad \text{for } i \in \mathcal{S}_t, \\ w_t^i = w_{t-1}^i, \quad \text{for } i \notin \mathcal{S}_t. \end{cases}$$

**Assumptions and notations.**

**A1.** *Uniform Lower Bound: There exists $f_* \in \mathbb{R}$ such that $f(x) \geq f_*$ for all $x \in \mathbb{R}^d$.*

**A2.** *Smooth Gradients: For any client $i$, the gradient $\nabla f_i(x)$ is $L$-Lipschitz continuous for some $L > 0$, i.e. for all $x, y \in \mathbb{R}^d$: $\|\nabla f_i(x) - \nabla f_i(y)\| \leq L\|x - y\|$.*

192 **A3.** *Bounded Variance: For any client $i$, the variance of the stochastic gradients is bounded by some*
193 $\sigma^2 > 0$, *i.e. for all* $x \in \mathbb{R}^d$: $\mathbb{E}[\|\widetilde{g}^i(x) - \nabla f_i(x)\|^2] \leq \sigma^2$.

194 **A4.** *Bounded Gradient Dissimilarity: There exist constants $G^2 \geq 0$ and $B^2 \geq 1$, such that for all*
195 $x \in \mathbb{R}^d$: $\sum_{i=1}^{n} \frac{\|\nabla f_i(x)\|^2}{n} \leq G^2 + B^2 \|\nabla f(x)\|^2$.

196 We define the notations required for the analysis. Consider a time step $t$, a client $i$, and a local step $q$.
197 We define

$$\mu_t = \left( w_t + \sum_{i=1}^{n} w_t^i \right) / (n+1)$$

198 the average over all node models in the system at a given time $t$. The first step of the proof is to
199 compute a preliminary upper bound on the divergence between the local models and their average.
200 For this purpose, we introduce the Lyapunov function: $\Phi_t = \|w_t - \mu_t\|^2 + \sum_{i=1}^{n} \|w_t^i - \mu_t\|^2$.

201 **Upper bounding the expected change in potential.** A key result from our analysis is to upper
202 bound the change (in expectation) of the aforementioned potential function $\Phi_t$:

203 **Lemma 2.** *For any time step $t > 0$ we have:*

$$\mathbb{E}\left[\Phi_{t+1}\right] \leq (1 - \kappa)\mathbb{E}\left[\Phi_t\right] + 3\frac{s^2}{n}\eta^2 \sum_{i=1}^{n} \mathbb{E}\left\|\check{h}_{t+1}^i\right\|^2, \quad \text{with } \kappa = \frac{1}{n}\left(\frac{s(n-s)}{2(n+1)(s+1)}\right).$$

204 The intuition behind Lemma 2 is that the potential function $\Phi_t$ remains concentrated around its mean,
205 apart from deviations induced by the local gradient steps. The full analysis involves many steps and
206 we refer the reader to Appendix B for complete proofs. In particular, Lemmas 16 and 18 allow us
207 to examine the scalar product between the expected node progress $\sum_{i=1}^{n} \check{h}_t^i$ and the true gradient
208 evaluated on the mean model $\nabla f(\mu_t)$. The next theorem allows us to compute an upper-bound
209 on the averaged norm-squared of the gradient, a standard quantity studied in nonconvex stochastic
210 optimization.

211 **Convergence results.** The following statement shows that `FAVAS` algorithm converges towards a
212 first-order stationary point, as $T$ the number of global epochs grows.

213 **Theorem 3.** *Assume A1 to A4 and assume that the learning rate $\eta$ satisfies $\eta \leq \frac{1}{20B^2bKLs}$. Then*
214 *FAVAS converges at rate:*

$$\frac{1}{T}\sum_{t=0}^{T-1}\mathbb{E}\|\nabla f(\mu_t)\|^2 \leq \frac{2(n+1)F}{Ts\eta} + \frac{Ls}{n+1}\left(\frac{\sigma^2}{n}\sum_{i}^{n}a^i + 8G^2b\right)\eta + L^2s^2\left(\frac{720\sigma^2}{n}\sum_{i}^{n}a^i + 5600bG^2\right)\eta^2,$$

215 *with $F := (f(\mu_0) - f_*)$, and*

$$\begin{cases} a^i, b = \frac{1}{\mathbf{P}(E_{t+1}^i>0)^2}\left(\frac{\mathbf{P}(E_{t+1}^i>0)}{K^2} + \mathbb{E}[\frac{\mathbb{1}(E_{t+1}^i>0)}{E_{t+1}^i \wedge K}]\right), \max_i\left(\frac{1}{\mathbf{P}(E_{t+1}^i>0)}\right) \textit{for } \alpha^i = \mathbf{P}(E_{t+1}^i>0)(E_{t+1}^i \wedge K), \\ a^i, b = \frac{1}{\mathbb{E}[E_{t+1}^i \wedge K]} + \frac{\mathbb{E}[(E_{t+1}^i \wedge K)^2]}{K^2\mathbb{E}[E_{t+1}^i \wedge K]}, \max_i\left(\frac{\mathbb{E}[(E_{t+1}^i \wedge K)^2]}{\mathbb{E}[E_{t+1}^i \wedge K]}\right) \textit{ for } \alpha^i = \mathbb{E}[E_{t+1}^i \wedge K]. \end{cases}$$

216 Note that the previous convergence result refers to the average model $\mu_t$. In practice, this does not
217 pose much of a problem. After training is complete, the server can ask each client to submit its final
218 model. It should be noted that each client communicates $\frac{sT}{n}$ times with the server during training.
219 Therefore, an additional round of data exchange represents only a small increase in the total amount
220 of data transmitted.

221 The bound in Theorem 3 contains 3 terms. The first term is standard for a general non-convex target
222 and expresses how initialization affects convergence. The second and third terms depend on the
223 statistical heterogeneity of the client distributions and the fluctuation of the minibatch gradients.
224 Table 1 compares complexity bounds along with synchronous and asynchronous methods.One can
225 note the importance of the ratio $\frac{s}{n}$. Compared to Nguyen et al. (2022) or Koloskova et al. (2022),
226 `FAVAS` can potentially suffer from delayed updates when $\frac{s}{n} \ll 1$, but `FAVAS` does *not* favor fast
227 clients at all. In practice, it is not a major shortcoming, and `FAVAS` is more robust to fast/slow clients
228 distribution than FedBuff/AsyncSGD (see Figure 2). We emphasize both FedBuff and AsyncSGD rely
229 on strong assumptions: neither the queuing process, nor the transitional regime are taken into account

in their analysis. In practice, during the first iterations, only fast clients contribute. It induces a serious bias. Our experiments indicate that a huge amount of server iterations has to be accomplished to reach the stationary regime. Still, under this regime, slow clients are contributing with delayed information. Nguyen et al. (2022); Koloskova et al. (2022) propose to uniformly bound this delay by some quantity $\tau_{max}$. We keep this notation while reporting complexity bounds in Table 1, but argue nothing guarantee $\tau_{max}$ is properly defined (i.e. finite). All analyses except that of Mishchenko et al. (2022) show that the number of updates required to achieve accuracy grows linearly with $\tau_{max}$, which can be very adverse. Specifically, suppose we have two parallel workers - a fast machine that takes only 1 unit of time to compute a stochastic gradient, and a slow machine that takes 1000 units of time. If we use these two machines to implement FedBuff/AsyncSGD, the gradient delay of the slow machine will be one thousand, because in the 1 unit of time we wait for the slow machine, the fast machine will produce one thousand updates. As a result, the analysis based on $\tau_{max}$ deteriorates by a factor of 1000.

In the literature, guarantees are most often expressed as a function of server steps. In the asynchronous case, this is *inappropriate* because a single step can take very different amounts of time depending on the method. For example, with FedAvg or Scaffold (Karimireddy et al., 2020), one must wait for the slowest client for each individual server step. Therefore, we introduce in Table 1 constants $C_-$ that depend on the client speed and represent the unit of time to wait between two consecutive server steps. Finally, optimizing the value of the learning rate $\eta$ with Lemma 12 yields the following:

**Corollary 4.** *Assume A1 to A4. We can optimize the learning rate by Lemma 12 and* `FAVAS` *reaches an $\epsilon$ precision for a number of server steps $T$ greater than (up to numerical constants):*

$$\frac{FL(\frac{\sigma^2}{n}\sum_i^n a^i + 8G^2 b)}{\epsilon^2} + (n+1)\left(\frac{FL^2(K^2\sigma^2 + L^2K^2G^2 + \frac{s^2\sigma^2}{n}\sum_i^n a^i + s^2G^2b)^{\frac{1}{2}}}{s\epsilon^{\frac{3}{2}}} + \frac{FB^2KLb}{\epsilon}\right),$$

*where $F = (f(\mu_0) - f_*)$, and $(a^i, b)$ are defined in Theorem 3.*

The second term in Corollary 4 is better than the one from the QuAFL analysis ($n^3$ of Zakerinia et al., 2022). Although this $(n + 1)$ term can be suboptimal, note that it is only present at second order from $\epsilon$ and therefore becomes negligible when $\epsilon$ goes to 0 (Lu and De Sa, 2020; Zakerinia et al., 2022).

**Remark 5.** *Our analysis can be extended to the case of quantized neural networks. The derived complexity bounds also hold for the case when the quantization function $Q$ is biased. We make only a weak assumption about $Q$ (we assume that there is a constant $r_d$ such that for any $x \in \mathbb{R}^d$ $\|Q(x) - x\|^2 \leq r_d$), which holds for standard quantization methods such as stochastic rounding and deterministic rounding. The only effect of quantization would be increased variance in the stochastic gradients. We need to add to the upper bound given in Theorem 3 an "error floor" of $12L^2 r_d$, which remains independent of the number of server epochs. For stochastic or deterministic rounding, $r_d = \Theta(d\frac{1}{2^{2b}})$, where $b$ is the number of bits used. The error bound is the cost of using quantization as part of the optimization algorithm. Previous works with quantized models also include error bounds (Li et al., 2017; Li and Sa, 2019).*

# 5 Numerical Results

We test `FAVAS` on three image classification tasks: MNIST (Deng, 2012), CIFAR-10 (Krizhevsky et al., 2009), and TinyImageNet (Le and Yang, 2015). For the MNIST and CIFAR-10 datasets, two training sets are considered: an IID and a non-IIID split. In the first case, the training images are randomly distributed among the $n$ clients. In the second case, each client takes two classes (out of the ten possible) without replacement. This process leads to heterogeneity among the clients.

The standard evaluation measure for FL is the number of server rounds of communication to achieve target accuracy. However, the time spent between two consecutive server steps can be very different for asynchronous and synchronous methods. Therefore, we compare different synchronous and asynchronous methods w.r.t. *total simulation time* (see below). We also measured the loss and accuracy of the model in terms of server steps and total local client steps (see Appendix C.3). In all experiments, we track the performance of each algorithm by evaluating the server model against an unseen validation dataset. We present the test accuracy and variance, defined as $\sum_{i=1}^{n} \|w_t^i - w_t\|^2$.

279 We decide to focus on non-uniform timing experiments as in Nguyen et al. (2022), and we base our
280 simulation environment on QuAFL's code[1]. After simulating $n$ clients, we randomly group them into
281 fast or slow nodes. We assume that at each time step $t$ (for the central server), a set of $s$ clients is
282 randomly selected without replacement. We assume that the clients have different computational
283 speeds, and refer to Appendix C.2 for more details. We assume that only one-third of the clients are
284 slow, unless otherwise noted. We compare FAVAS with the classic synchronous approach FedAvg
285 (McMahan et al., 2017) and two newer asynchronous metods QuAFL (Zakerinia et al., 2022) and
286 FedBuff (Nguyen et al., 2022). Details on implementing other methods can be found in Appendix C.1.

287 We use the standard data augmentations and normalizations for all methods. FAVAS is implemented in
288 Pytorch, and experiments are performed on an NVIDIA Tesla-P100 GPU. Standard multiclass cross
289 entropy loss is used for all experiments. All models are fine-tuned with $n = 100$ clients, $K = 20$
290 local epochs, and a batch of size $128$. Following the guidelines of Nguyen et al. (2022), the buffer
291 size in FedBuff is set to $Z = 10$. In FedAvg, the total simulated time depends on the maximum
292 number of local steps $K$ and the slowest client runtime, so it is proportional to the number of local
293 steps and the number of global steps. In QuAFL and FAVAS on the other hand, each global step has a
294 predefined duration that depends on the central server clock. Therefore, the global steps have similar
295 durations and the total simulated time is the sum of the durations of the global steps. In FedBuff, a
296 global step requires filling a buffer of size $Z$. Consequently, both the duration of a global step and
297 the total simulated time depend on $Z$ and on the proportion of slow clients (see Appendix C.2 for a
298 detailed discussion).

299 We first report the accuracy of a shallow neural network trained on MNIST. The learning rate is set
300 to $0.5$ and the total simulated time is set to $5000$. We also compare the accuracy of a Resnet20 (He
301 et al., 2016) with the CIFAR-10 dataset (Krizhevsky et al., 2009), which consists of 50000 training
302 images and 10000 test images (in 10 classes). For CIFAR-10, the learning rate is set to $0.005$ and the
total simulation time is set to $10000$. In Figure 1, we show the test accuracy of FAVAS and competing

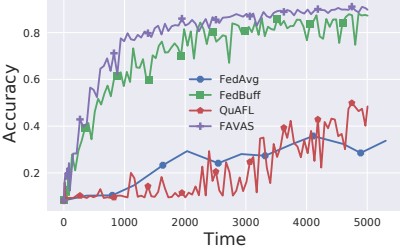

Figure 1: Test accuracy on the MNIST dataset with a non-IID split in between $n = 100$ total nodes, $s = 20$.

Table 2: Final accuracy on the test set (average and standard deviation over 10 random experiments) for the MNIST classification task. The last two columns correspond to Figures 1 and 2.

| Methods | IID split | non-IID split ($\frac{2}{3}$ fast clients) | non-IID split ($\frac{1}{9}$ fast clients) |
|---|---|---|---|
| FedAvg | $93.4 \pm 0.3$ | $38.7 \pm 7.7$ | $44.8 \pm 6.9$ |
| QuAFL | $92.3 \pm 0.9$ | $40.7 \pm 6.7$ | $45.5 \pm 4.0$ |
| FedBuff | $\mathbf{96.0} \pm 0.1$ | $85.1 \pm 3.2$ | $67.3 \pm 5.5$ |
| FAVAS | $95.1 \pm 0.1$ | $\mathbf{88.9} \pm 0.9$ | $\mathbf{87.3} \pm 2.3$ |

303
304 methods on the MNIST dataset. We find that FAVAS and other asynchronous methods can offer a
305 significant advantage over FedAvg when time is taken into account. However, QuAFL does not
306 appear to be adapted to the non-IID environment. We identified client-side updating as a major
307 shortcoming. While this is not severe when each client optimizes (almost) the same function, the
308 QuAFL mechanism suffers from significant client drift when there is greater heterogeneity between
309 clients. FedBuff is efficient when the number of stragglers is negligible compared to $n$. However,
310 FedBuff is sensitive to the fraction of slow clients and may get stuck if the majority of clients are
311 classified as slow and a few are classified as fast. In fact, fast clients will mainly feed the buffer,
312 so the central updates will be heavily biased towards fast clients, and little information from slow
313 clients will be considered. Figure 2 illustrates this phenomenon, where one-ninth of the clients are
314 classified as fast. To provide a fair comparison, Table 2 gives the average performance of 10 random
315 experiments with the different methods on the test set.

316 In Figure 3a, we report accuracy on a non-IID split of the CIFAR-10 dataset. FedBuff and FAVAS
317 both perform better than other approaches, but FedBuff suffers from greater variance. We explain
318 this limitation by the bias FedBuff provides in favor of fast clients. We also tested FAVAS on the
319 TinyImageNet dataset (Le and Yang, 2015) with a ResNet18. TinyImageNet has 200 classes and each

[1] https://github.com/ShayanTalaei/QuAFL

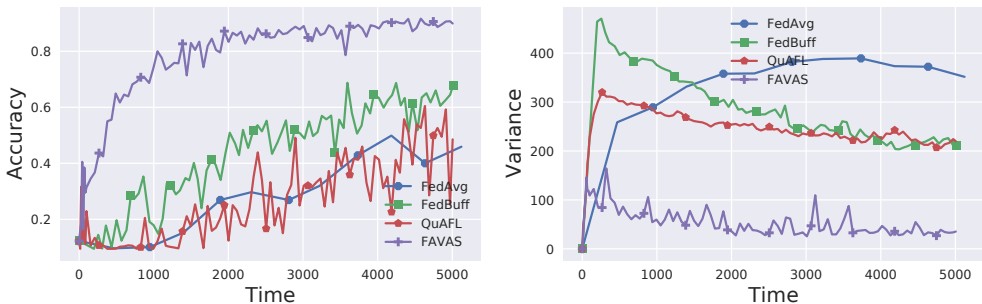

Figure 2: Test accuracy and variance on the MNIST dataset with a non-IID split between $n = 100$ total nodes. In this particular experiment, one-ninth of the clients are defined as fast.

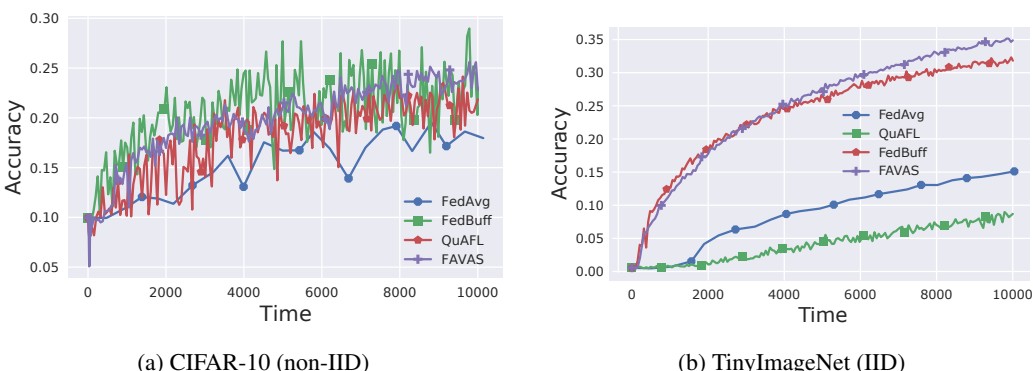

(a) CIFAR-10 (non-IID)

(b) TinyImageNet (IID)

Figure 3: Test accuracy on CIFAR-10 and TinyImageNet datasets with $n = 100$ total nodes. Central server selects $s = 20$ clients at each round.

class has 500 (RGB) training images, 50 validation images and 50 test images. To train ResNet18, we follow the usual practices for training NNs: we resize the input images to $64 \times 64$ and then randomly flip them horizontally during training. During testing, we center-crop them to the appropriate size. The learning rate is set to $0.1$ and the total simulated time is set to $10000$. Figure 3b illustrates the performance of FAVAS in this experimental setup. While the partitioning of the training dataset follows an IID strategy, TinyImageNet provides enough diversity to challenge federated learning algorithms. Figure 3b shows that FAVAS scales much better on large image classification tasks than any of the methods we considered.

**Remark 6.** *We also evaluated the performance of FAVAS with and without quantization. We ran the code [2] from LUQ (Chmiel et al., 2021) and adapted it to our datasets and the FL framework. Even when the weights and activation functions are highly quantized, the results are close to their full precision counterpart (see Figure 7 in Appendix C).*

## 6 Conclusion

We have presented FAVAS the first (centralised) Federated Learning method of federated averaging that accounts for asynchrony in resource-constrained environments. We established complexity bounds under verifiable assumptions with explicit dependence on all relevant constants. Empirical evaluation shows that FAVAS is more efficient than synchronous and asynchronous state-of-the-art mechanisms in standard CNN training benchmarks for image classification.

---

[2]https://openreview.net/forum?id=clwYez4n8e8

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
