# OpenReview forum: "FAVAS: Federated AVeraging with ASynchronous clients"
_NeurIPS.cc/2023/Conference — Submitted to NeurIPS 2023_

### Official Review · Reviewer_PbVJ · 2023-07-04

**Soundness:** 3 good
**Presentation:** 3 good
**Contribution:** 2 fair
**Rating:** 5
**Confidence:** 3

**Summary:**

The authors propose a new Asynchronous Federated Learning algorithm, FAVAS. They provide theoretical convergence analysis of the proposed algorithm as well as present experimental results. The main motivation behind their framework is to handle heterogeneities in the computation speed of clients in asynchronous settings. The work’s fundamental contributions are: 1) Compared to the algorithm proposed in QuAFL (Zakerinia et al., 2022), FAVAS weights updates of each contacted client according to their number of local iterates which is allowed to be different across clients. Its asymptotic bound on the total number of clients (n) is improved. 2) Authors present that the proposed algorithm is experimentally better than baselines on several datasets.

**Strengths:**

1. The proposed algorithm seems efficient in terms of the utilization of clients. In conventional FL analysis, clients generally start computations when they are interrupted by the server. Here, clients always do computation (until $K$ local steps) and send the update when they are interrupted.

2. Although the algorithm is similar to QuAFL (Zakerinia et al., 2022), the authors redesigned local updates and local weighting so that aggregation becomes unbiased. I like how they highlighted the differences in Algorithm 1.

3. The mathematical analysis of the convergence seems strong and doesn’t use any unnatural assumptions. The intuition behind the proof abstract is well explained. In particular, I like the explanations in the main paper while the full proof is presented in Appendix.

4. Both in algorithmic and experimental analysis, the authors provide a good comparison with baselines in the literature. The authors successfully present their experimental setup details.

**Weaknesses:**

1. A discussion on the relation between the number of local iterates and training quality can enrich the study. In the proposed framework, the number of local iterates that slow & fast clients take are random and depend on the server interaction time and clients' computation time. On the contrary, in most baseline methods, the number of local iterates is a parameter that we set and directly affects the training time. For a fair comparison with baselines, some information (maybe the mean and standard deviation statistics) on the number of local iterates during training may be helpful.

2. Lack of novelty: As the authors stated, the proof strategy and the algorithm are inspired by QuAFL (Zakerinia et al., 2022). I suspect the theoretical novelty in this work is above the threshold for the conference requirement.

**Questions:**

1. Line 42: Which question do authors refer to? The sentence may be changed, or the expression may be clarified.

2. Line 45: The term “unbiased aggregation” can be confusing. Multiple factors can create bias in aggregation (staleness bias, client selection bias, bias in the number of local updates across clients, etc.). This can be clarified.

3. Line 50: The authors claim that the algorithm's guarantee does not depend on the maximum delay. However, I think there should still be an assumption about client delays. For example, if all clients are too slow to compute even one local update before server interaction, the algorithm seems to stop. It would be good if the authors could very briefly discuss this (for reference, please see Discussion in Section 3.2 Algorithm Description of QuAFL (Zakerinia et al., 2022).)

4. Line 97: In QuAFL (Zakerinia et al., 2022), I think that each client doesn’t have to compute $K$ local updates. The just send the progress taken since last contacted. For instance, “Specifically, in the analysis we treat the number of local steps as a random variable which takes values in {1, 2, ...K} and define its expected value to be H. Intuitively, a larger H gives better convergence, since this means that nodes are able to perform larger number of local steps.” part can be found in their paper. Can authors clarify what they mean?

Also, it is mentioned in Lines 142, 179 and 440. Although the authors said “QuAFL assume the same number (K) of local iterates in analysis,” I believe it is not the case in QuAFL (e.g. see appendix A.1 of their paper). They consider the number of local iterates as random variables.

5. Line 119: Authors said, “Only Mishchenko et al. (2022) analyze the asynchronous SGD independently of the delays in the gradients.” I remembered QuAFL convergence doesn’t depend on delays either. Could you explain this?

6. In Table 1, C_ terms seem vague to me. Although it is said to be a “constant” value for the time between two consecutive iterates, I think it is more correct to say “average” round time in the whole training process due to randomness.

7. Lines 216-220: Although it may not be a problem in practice, I couldn’t get the authors’ claim here. The authors stated that an additional round after training would be enough. However, the convergence analysis does not bound $‖\nabla f(\mu_T)‖$, the final version of average models. The analysis proves that the average of all $\mu_t$ s over $t=1,\dots,T$ has a bounded gradient norm. So, to get a model whose gradient norm is bounded, we would need all $\mu_1,\dots,\mu_T$. Therefore, we would need much more communication to access all $\mu_1,\dots,\mu_T$. Could authors clarify this?

8. Line 227: Authors stated that FAVAS doesn’t favor fast clients compared to Koloskova et al. (2022). In Koloskova et al. (2022), the client selection happens uniformly at random in their Algorithm 2. So, I am not sure why the authors think that it favors fast clients. On the other hand, I have found authors right in the same claim for comparison with Nguyen et al. (2022).

9. Line 458: Is $\mathbb{E}[Y_q]$ a typo that should be $\mathbb{E}[Y^q]$?

**Limitations:**

Please see question 3 in Questions. The authors discussed both theoretical and experimental extends of the study adequately.

The authors also discussed the “Environmental footprint” of their study. The reviewer thanks them for their responsible attitude towards our environment.

---

> ### Author Rebuttal · Authors · 2023-08-04
>
> We thank the reviewer for his careful reading and comments. We are pleased that the reviewer appreciates that the mathematical analysis "does not use unnatural assumptions", and that our experimental results "provide a good comparison with baselines in the literature." We also sincerely thank the reviewer for his comment on our ecological footprint. We respond to each comment as follows.
>
> Weakness 1 "information on the number of local iterates during training may be helpful" In the following Table we present some statistics about the number of local epochs computed while training FAVAS on MNIST classification task with the set of hyperparameters detailed in the paper.
> | mean $\pm$ std (slow clients) | mean $\pm$ std (fast clients) |
> | -------- | ------- |
> | $1.2 \pm 2.0$| $11.8 \pm 7.1$    |
>
> Weakness 2 "Lack of novelty in the proof" We want to highlight our contribution on the analysis part is two fold. First, as the local update strategy we propose (line.26 in Algo.1) is different from QuAFL's, the "contraction Lemma" (line.203) is different from QuAFL's, and our rate is better. Second, we want to *emphasize* that we do not assume clients do the same (on average) number of local steps, what QuAFL does, and we propose a new reweighting strategy with $\alpha_i$. Note the authors of QuAFL have provided on arxiv a new version (v3) in which they get rid of this assumption, and follow our setting (the paragraph "the number of local steps taken by client i between server interactions...the individual step distributions $H_i$ can be completely different." were added page.5 in v3 after the conference deadline).
>
> Q.1 "Line 42: Which question do authors refer to?" We agree the formulation is misleading. We refer to the "challenge" we describe line.35. We will reformulate this sentence.
>
> Q.2 "unbiased aggregation can be confusing" We definitely agree, biasedness can come from different factors. We will explicitly right "aggregation that does not favor fast client" instead of "unbiased aggregation”.
>
> Q.3 " if all clients are too slow...the algorithm seems to stop" We agree the update frequency (tuned by the central server) must be set to avoid that all clients dont't have time to compute at least one gradient estimation. In practice (for $n$ large enough) it is very unlikely that all clients are too slow. QuAFL (v3 section.3.2 page.9) also mention it "Otherwise, the sampling frequency of the server is too high, and the server should simply decrease its frequency". We will add a discussion about it in the main text.
>
> Q.4 "Line 97...client doesn’t have to compute local updates" We agree clients in QuAFL dont't make exactly $K$ local steps. We say line.97 that " Each client must compute K local steps and can be interrupted by the central server at any time.". It means that in QuAFL, each client receives a model and a maximum number of local epochs $K$, and in practice it computes a number of local epochs in between $0$ and $K$.
>
> Q.about QuAFL assumption "Also, it is mentioned in Lines 142, 179 and 440...local iterates as random variables." We definitely agree QuAFL's authors consider the number of local steps as a random variable. But we want to highlight that QuAFL considers all clients compute the *same* number of local epochs on average: i.e. clients share the same expected value. Note this was fixed in their new version v3 in arxiv (but after the conference deadline). In lines.142, 179 and 440 we do not argue that QuAFL assume the same number of local iterates exactly equal to the limit $K$. We will rephrase it to make it clearer as the following "It is assumed that clients compute the same number of local epochs on average in the analysis from Zakerinia et al. (2022)".
>
> Q.5 "QuAFL convergence doesn’t depend on delays either" We agree QuAFL is the only work in addition to Mishchenko et al. (2022) that provides an analysis that does not depend on some $\tau_{max}$ quantity. But, to be fair, QuAFL makes use of an additional assumption (all clients compute the same number of local epochs on average) that is infeasible in practice. We will change the sentence line.119 to incorporate QuAFL (and its additional assumption).
>
> Q.6 "C_ terms seem vague to me" We agree this terms are average value that depends on the distribution of client speeds, rather than constant terms. It is difficult to have a closed form of the aforementioned term (one can obtain an estimation in the limit of the stationary regime under assumptions upon the distribution of client speeds). We will rephrase the caption of Table.1 to avoid any mistakes.
>
> Q.7 "the convergence analysis does not bound $\mu_T$" We are happy that you underline that our bounds concerns the average (wrt time) of models rather than the ultimate model. But this is usual in the non-convex literature: analysis provide some bounds on the average of the gradient of all the model between step $0$ and $T$, never on the final iterate. In practice people do *not* "get a model whose gradient norm is bounded", they just run the optimization process and keep the last iterate. We do the same: we have some guarantees on the average of gradient models, but we stick to the ultimate iterate in practice.
>
> Q.8 "I am not sure why the authors think that it favors fast clients." We agree AsyncSGD from Koloskova et al. (2022) samples clients uniformly at random. This results to equal contribution of clients under the stationary regime: fast clients are not favored. However, during the transient regime, fast clients are mainly contributing. In AsyncSGD, slow clients will be stacking tasks (gradient computations), and contribute sparingly during the first server steps. Only when the stationary regime is reached, slow clients will contribute as much as fast clients, on average.
>
> Q.9 Thank you for the typo ! We will fix it.
>
> We thank the reviewer again for his/her comments and support. We would be happy to provide more elements if there remain any unresolved questions.

---

> > ### Comment · Reviewer_PbVJ · 2023-08-17
> > **Thanks for the response!**
> >
> > I've carefully read all the reviews and comments. I sincerely appreciate authors for their detailed responses. They have substantially adressed my concerns.
> >
> > About QuAFL's arXiv submission, you are right. There are three versions on arXiv.  In the last version, it seems that the authors removed the necessity that clients compute the same number of local epochs on average. (You can see that $H_i$ defined in their paper can be different across clients ($i$ is the client index)) You may want to mention it as a seperate work done concurrently.
> >
> > My minor concerns remaining are:
> >
> > - Q7. You are right in what you stated in your response. In practice it doesn't make a difference. Also, in the non-convex literature, it is general to bound $\frac{1}{T}\sum_{t=0}^{T-1}||x_t||^2$. But it still doesn't answer my question. I suspect Lines 216-200 may be misleading in this sense: Lines 216-220 seem suggesting that "although the convergece result refers to the average of average model $\gamma_t$ over time, it matches with the practice. We will need just one more round (to get $\gamma_{T-1}$) to have a sequence whose convergence is guaranteed." However, as you also said, we do not know the convergence of $\gamma_{T-1}$ but $\frac{1}{T}\sum_{t=0}^{T-1}||\gamma_t||^2$. So, in your algorithm, even if we do this one more round in the end, we cannot have a model whose convergence is guaranteed. As I said, that is fine for all practical senses or me. I just wanted to clarify. You may consider adding a sentence like "In practice, this does not pose much of a problem since just the final version of the trained model is considered."
> >
> > I have one more additional concern:
> > Could you clarify why the asynchrony is needed if we let clients do their local training at their own speed? The convergence of synchronous FL with varying number of local iterates is shown in [1]. Then why do we need client-sampling? Why does setting a round deadline and calling all clients' updates so far in that round not work? I mean, why don't we set $s$ to $n$?
> >
> >
> > [1] Wang, J., Liu, Q., Liang, H., Joshi, G., and Poor, H. V. (2020). Tackling the objective inconsistency problem in heterogeneous federated optimization. Advances in neural information processing systems, 33, 7611-7623.

---

> > > ### Author Response · Authors · 2023-08-17
> > >
> > > We thank again the reviewer for his/her time and insightful comments.
> > >
> > > 1) "About QuAFL's arXiv submission" Thank you for your feedback. We will mention it as a separate work done concurrently in our main text.
> > >
> > > 2) Thank you for your clarification about Q.7. Following your advice, we will add a sentence to explain that in practice we consider the final version of the trained model.
> > >
> > > 3) New Question: We do agree that full concurrency ($s=n$) with asynchronous clients is equivalent to the synchronous framework considered in [1]. But it is not true as long as $s<n$, because in this case clients can work on potentially outdated models. Thus, we finally get to your interesting question: "Then why do we need client-sampling?". Consider a practical network with an important number of clients ($n=10^6$ for e.g.), one would require a very large bandwidth to apply any methods that involves full concurrency (in particular for the upload phase). Federated optimization also comes along some limitations/properties. In particular, for communication (see FedAvg [2] page.2), one can consider "mobile devices are frequently offline or on slow or expensive connections". In addition, for homogeneous loss functions, FedAvg [2] (page.6) report that there is no advantage in increasing $s$. This explains why client-sampling is necessary in a realistic FL scenario.
> > >
> > > We hope that these explanations will address your concerns. Thank you again for your support.
> > >
> > >
> > > [1] Wang, J., Liu, Q., Liang, H., Joshi, G., and Poor, H. V. (2020). Tackling the objective inconsistency problem in heterogeneous federated optimization. Advances in neural information processing systems, 33, 7611-7623.
> > >
> > > [2] McMahan, B., Moore, E., Ramage, D., Hampson, S., and y Arcas, B. A. (2017). Communication-efficient learning of deep networks from decentralized data. In Artificial intelligence and statistics, pages 1273–1282. PMLR

---

> > > > ### Comment · Reviewer_PbVJ · 2023-08-17
> > > >
> > > > Thanks for your quick response! Most of my concerns are resolved now. I am increasing my score from 4 to 5.

---

### Official Review · Reviewer_GoYN · 2023-07-06

**Soundness:** 3 good
**Presentation:** 2 fair
**Contribution:** 2 fair
**Rating:** 4
**Confidence:** 4

**Summary:**

This paper proposes a new asynchronous federated learning update scheme to deal with the biasedness of conventional asynchronous updates which favor fast clients. The proposed method involves two main contributions, one is the direct local model update and the other is to reweight the contributions for the selected clients. The authors provide theoretical convergence analysis for the proposed method under stochastic non-convex settings. The numerical experimental results of the proposed method outperform other baselines in most settings.


**Strengths:**

1. The paper is well structured with clear motivations. The theoretical analysis and numerical results are clear to support the intuition.

2. The analysis of the proposed method can get rid of the constraint of bounded gradient and the uniformly bounded delay, which among the analysis of papers studying asynchronous FL, only very few of them do not require both conditions.


**Weaknesses:**

1. Some key information about the reweighted scheme need some further illustration.

2. In my opinion, one difference between the proposed method and some previous baselines such as FedBuff and FedAsync is that the proposed FAVAS enables different clients to perform unfix steps of local training during two communication rounds, and the reweighted scheme is proposed to overcome the biasedness of unfix local steps. The independent number of local update steps to perform in each round based on the communication status shares a similar idea as [1]. Thus, the idea of FAVAS is similar to combining the QuAFL and [1] into a single framework.

3. One related paper [1] about federated learning could be discussed.

[1] Yang, H., Zhang, X., Khanduri, P., & Liu, J. (2022, June). Anarchic federated learning. In International Conference on Machine Learning (pp. 25331-25363). PMLR.

**Questions:**

1. I am still quite confused about the reweight parameter $\alpha^i$ and the parameter $E_{t+1}^i$ in Lines 149-150. How does $E_{t+1}^i$ be defined? Is $E_{t+1}^i$ a tunable parameter or a predicted value, as in Line 177 “Each random variable $E_t^i$ models the number”, it seems a predicted one, thus how to predict it.

2. What do the parameters $a^i$ and $b$ in Theorem 3 stand for? It seems they are highly related to the reweighted parameters.

3. Regarding the convergence rate, it is sure that FAVAS obtains a better rate for the non-dominant term than QuAFL. However, can you show a detailed comparison between the rate of QuAFL and FAVAS in Table 1 (especially for the dominant $O(\epsilon^{-2}$ term)?

4. Could you provide some details about the choosing of reweight parameter $\alpha^i$ and parameter $E_{t+1}^i$ for the numerical results?



**Limitations:**

see above

---

> ### Author Rebuttal · Authors · 2023-08-04
>
> We thank the reviewer for his time and insightful comments. We also thank the reviewer for highlighting that our paper "is well structured with clear motivations". We respond to each comment below.
>
> Weaknesses 2,3 : Thank you for the reference about Anarchic Federated Learning [1], we will add a discussion in the related works section. First, one can note all bounds in [1] depend on some maximum delay $\tau_{max}^2$. This quantity is not a well-defined - because under realistic assumptions of speed distribution, the delay has no reason to be bounded (see additional pdf), and seriously damages the complexity bounds (see our discussion in line.234-242). We agree the idea from [1] (line.3 in Algo.2 "Sum and rescale the stochastic gradients") is similar to our reweighting strategy with $\alpha_i$. However [1] does not prescribe any rule to fix the value of the local steps "$K_t^i$", and let them freely appear in the convergence rate. Whereas in FAVAS we do provide 2 different methods to compute the $\alpha_i$, given that you only have access to the client speeds distribution $E_t^i$. In appendix C.2, we assume that the clients have different computational speeds that result in $E_t^i$ distributed according to a geometrical distribution of parameter $\lambda^i$: $E^i_t \sim$ $\operatorname{Geom}(\lambda^i)$. The parameter $\lambda^i$ is $1 / 2$ for fast clients and $1 / 16$ for slow clients; the expected local steps $\mathbb{E}[E^i_t]$ is 2 and 16, respectively.
>
> Q.1 "How does $E_t^i$ be defined? " : Clients evolve freely and can get interrupted at any moment by the central server. Hence we can assume the number of local epochs a client computes follow a geometric distribution (Poisson if we would consider the continuous case). We want to rescale this number of locally computed epochs by some $\alpha_i$ that preserves the unbiasedness property *and* only depends on client speeds. Either we rescale the sum of gradients by the exact number of locally computed epochs (the random variable $E^i_{t+1} \wedge K$), or we rescale it by the expected value of the number of local epochs ($\mathbb{E}[E^i_{t+1} \wedge K]$).
>
> Q.2 "What do the parameters $a_i$ and $b$ in Theorem 3 stand for? " : The exact value of this terms depends on $\alpha_i$, and is given line.215. This terms indicate how much the variance of the gradients ($\sigma^2$ and $G^2$) impact the optimization.
>
> Q.3 "Regarding the convergence rate, it is sure that FAVAS obtains a better rate for the non-dominant term than QuAFL. However, can you show a detailed comparison between the rate of QuAFL and FAVAS" : We agree it is not easy to fairly compare complexity terms in Table.1, especially because QuAFL makes an additional assumption to ease the calculation. Let's also assume that each client has the same speed (on average) and run the same number (on average) of local steps $E$. Take $K$ large enough wrt $E$ and $G=0$ (homogeneous loss functions) to simplify. Under these assumptions, $\sum\nolimits_{i=1}^n\frac{a_i}{n}$ is about $\frac{1}{E}$ ($+ o(\frac{1}{K^2}$)). Hence, $\sum\nolimits_{i=1}^n\frac{a_i}{n} \ll \frac{K}{E^2}$. As a consequence, the leading term (in $\epsilon^{-2}$) from FAVAS ($FL\sigma^2\sum\nolimits_{i=1}^n\frac{a_i}{n}$) is smaller than the leading term from QuAFL ($FL\sigma^2\frac{K}{E^2}$).
>
> Q.4 "Could you provide some details about the choosing of reweight parameter $\alpha_i$ and parameter $E_t^i$ for the numerical results?" : First, $E_t^i$ is not a parameter we can choose, but a random variable that is directly linked to the distribution of client speeds. In appendix C.2, we assume that the clients have different computational speeds that result in $E_t^i$ distributed according to a geometrical distribution of parameter $\lambda^i$: $E^i_t \sim$ $\operatorname{Geom}(\lambda^i)$. The parameter $\lambda^i$ is $1 / 2$ for fast clients and $1 / 16$ for slow clients; the expected local steps $\mathbb{E}[E^i_t]$ is $2$ and $16$, respectively. Second, we propose two versions (see line.181) for setting $\alpha_i$: a stochastic one and a deterministic one. For the stochastic one, $\alpha_i$ is a random variable: it is the (random) number of local steps a client has time to compute before getting interrupted by the central server. In the code (provided in supplementary materials), this can be found line.349 ("initial_client_dict[key]   = (1/taken_steps)*client_dict[key] + (1-1/taken_steps)*initial_client_dict[key]") in the file "\favas_supplementary_neurips_2023\favas_code_neurips_2023\trainer.py". For the determenistic one, $\alpha_i$ is a constant (expected value) that only depends on client speed. In the code provided, this can be found line.400 ("client.mean_step_time") in the file "\favas_supplementary_neurips_2023\favas_code_neurips_2023\trainer.py".
>
> We thank the reviewer again for his/her comments and support. We would be happy to provide more elements if there remain any unresolved questions.
>
> [1] Yang, H., Zhang, X., Khanduri, P., & Liu, J. (2022, June). Anarchic federated learning. In International Conference on Machine Learning (pp. 25331-25363). PMLR.

---

> > ### Comment · Reviewer_GoYN · 2023-08-20
> >
> > I thank the authors for the response. While most of the concerns are answered, I remain a bit concerned about the technical novelty of the proposed work for its similarity to QuAFL and AnarchicFL.
> >
> > After reading other reviewers' comments. I also have a new question on the asynchronization mechanism. Asynchronized algorithms are proposed to avoid the waiting caused by slow stragglers: even if one client takes forever to return the results, the overall training procedure will not need to wait for it.
> >
> > The proposed "asynchronization" seems to work by allowing different clients to have different numbers of local steps. And in this case, it seems the waiting can still happen here: if one client were selected in the current round and stopped training since it was contacted by the server but stuck by a very low communication bandwidth, it seems all other client/server will need to wait until this client finishes uploading the update before moving on to the next round. In this sense, this is not a fully asynchronized algorithm.
> >
> > Can the authors clarify on this issue? Thanks

---

### Official Review · Reviewer_Aa1R · 2023-07-06

**Soundness:** 4 excellent
**Presentation:** 3 good
**Contribution:** 3 good
**Rating:** 7
**Confidence:** 4

**Summary:**

Authors introduce a novel asynchronous centralized Federated Learning (FL) algorithm which corrects for clients with varying computational speeds (and delays). Theoretical convergence guarantees are provided as well as strong empirical results. Solves an interesting issue of model degradation when faster clients dominate asynchronous updates.

**Strengths:**

1. Well-written paper with nice tables and clear algorithms.
2. Strong theoretical backing which align with peer algorithms (without favoring faster clients).
3. Impressive empirical results which showcase the algorithms importance in realistic (heterogeneous) settings for FL.

**Weaknesses:**

1. Novelty of paper seems to lie solely in the reweighting of fast/slow clients. This is important but seems to be the only main contribution. Algorithm is extremely close to QuAFL otherwise.
2. Convergence is theoretically degraded when a large subset of devices are sampled to participate (s/n ≪ 1). This is a cause for concern in realistic applications.

**Questions:**

One small theoretical piece of feedback. The theory incorporates a Bounded Gradient Dissimilarity (BGD) assumption. This can likely be improved upon (to boost non-iid results) by incorporating a gradient tracking scheme. Algorithms such as BEER [R1] and D2 [R2] (not my papers I swear!) are able to remove non-iid assumptions and empirically improve their performance compared to peer algorithms by incorporating gradient tracking (utilizing consecutive gradient updates). Maybe adding this in can further boost your empirical results!


It seems to me that the main idea is to reweight slow and fast worker models so that they're equal when performing aggregation in the asynchroous setting. General question: is favoring faster clients that big of an issue? Slow clients would be contributing a model that has barely been trained and may drag the consensus model to a bad local optima. Based on your empirical results my intuition is wrong, but it would be nice to gain an explanation why reweighting slow clients is important.


As mentioned in the paper, "FAVAS can potentially suffer from delayed updates when s/n ≪ 1,". This could be problematic in many realistic implementations of the algorithm where only a small fraction of devices can partake in each round. Where does this s/n term come from and is it an artifcat of the theory or can it be removed?

Why does FedAvg have the largest variance? Isn't it a synchronous method? Shouldn't there should be 0 variance?

Is it possible to gain intuition why FedBuff and FAVAS perform well in the non-iid setting? The non-iid setting is very difficult, and I haven't seen many instances where test accuracies are above 80% for CIFAR10.

Small note: keep colors consistent for the algorithms in the plots.

[R1] Zhao, Haoyu, et al. "BEER: Fast $O (1/T)$ Rate for Decentralized Nonconvex Optimization with Communication Compression." Advances in Neural Information Processing Systems 35 (2022): 31653-31667.

[R2] Tang, Hanlin, et al. "$D^ 2$: Decentralized training over decentralized data." International Conference on Machine Learning. PMLR, 2018.

**Limitations:**

Limitations are not explicitly outlined in a section. I would recommend adding into the appendix at a minimum.

---

> ### Author Rebuttal · Authors · 2023-08-04
>
> We thank the reviewer for his/her time, careful reading, and insightful comments, and for finding that our paper is "well-written with nice tables and clear algorithms".
> We answer to each comment afterwards.
>
> Q.1 "...incorporating a gradient tracking scheme" Thank you for the interesting references! By now we already tried to adapt the method from Scaffold [1] to our setting and we can incorporate some control variates. But when in Scaffold one
> has delays only on the control variates, in the asynchronous setting, one has delays on control variates and on the gradient evaluations. This prevent us from obtaining a control variate drift that is contractive. We will have a deeper look at the references about gradient tracking, thank you !
>
> Q.2 "is favoring faster clients that big of an issue?" This is a very important question and the answer depends on the setting one considers. If client evaluates homogeneous functions, one can favor fast clients. However it is not true when we consider clients with heterogeneous (loss) functions. As a toy example, consider we are working on MNIST with only 2 clients: the fast one has only access to digits 0 to 4, and the slow one has access to the digits 5 to 9. If we favor the fast client (as FedBuff or QuAFL would), the central server model will be very good at classifying digits from 0 to 4, but very bad at classifying digits from 5 to 9. Hence the final central server accuracy on the test set will be around $50$%. However, if we reweight (i.e., give more importance to contributions from slow client, and less weight to contributions from fast client) the clients, the central server model will be able to better classify all digits. We agree "slow clients would be contributing a model that has barely been trained and may drag the consensus model to a bad local optima", and it explains why we can not reach a $99$% on MNIST: slow clients contributions have a lower quality compared to fast clients ones (because slow clients will be able to compute less local steps). But it is better to take into consideration all clients contributions (whether there are "bad or good" quality ones), rather than only focusing on "good" contributions from fast clients only.
>
> Q.3 "Where does this s/n term come from and is it an artifact of the theory or can it be removed?" First note this s/n term is not critical in our analysis, as it only concerns second order terms. Second, this term is much lower compared to other asynchronous baselines FedBuff and QuAFL. Consider an example where $s=1$ client is selected at each round, $n=10^{4}$ total clients are available on the network, the network suffer from a maximum delay of $\tau_{max} = 10^{3}$ (note this quantity is in reality infinite and grows with the number of server steps; see additional pdf), and the distribution of client speeds results into an average units of time between two consecutive server steps about $C_{FedBuff}=10$. Under this setting, the second order term from FAVAS will scale as $\simeq \frac{n}{s} = 10^{4}$, whereas it will scale as $\simeq \frac{n\sqrt{n}}{\sqrt{s}}=10^{6}$ for QuAFL, and it will scale as $\simeq {\tau_{max}^2 C_{FedBuff}}=10^{7}$ for FedBuff.
>
> Q.4 "Why does FedAvg have the largest variance? Isn't it a synchronous method? Shouldn't there should be 0 variance?" This is an interesting remark. We plot the variance between the central server model and the local client models that are optimized. We agree with you, the variance of FedAvg is 0 initially because all clients start the training process from the same point which is exactly the central server model. However the local model computed by some client $i$ can diverge (the so-called client drift phenomenon) from its starting point. This explains the variance in the models for FedAvg. This variance is small for FAVAS because either slow clients only have computed few steps (and thus can not suffer from important client drift), or because fast clients have their gradients rescaled by an important value (the weight $\alpha_i$ prevents them from getting too far from the starting point). We will add a discussion in the main text about it.
>
> Q.5 "Is it possible to gain intuition why FedBuff and FAVAS perform well in the non-iid setting?..." : The non-iid setting performances highly depend on how strong the "non-iidness" is. In our setting, each client has only access to 2 classes, this is a very heterogeneous setting, and the optimization is very complicated. In the literature you can find some "gentle non-iidness" where the dataset is split with Dirichlet with small parameters, and one can recover decent performances (above $70$%) even on CIFAR-10. But we believe our setting is closer to the reality (think about a network of hospitals where patients are split wrt age). FedBuff and FAVAS perform better in the non-iid setting because they aggregate contributions from different clients at the same time and with the same weight (what implicitly QuAFL does not).
>
> We thank the reviewer again for his/her numerous questions, and hope our answers clarified his/her concerns. We would be happy to provide more elements if there remain any unresolved questions.
>
> [1] Karimireddy, S. P., Kale, S., Mohri, M., Reddi, S., Stich, S., and Suresh, A. T. Scaffold: Stochastic controlled averaging for federated learning. In International Conference on Machine Learning, pp. 5132–5143. PMLR, 2020.

---

> > ### Comment · Reviewer_Aa1R · 2023-08-18
> >
> > I appreciate the thoughtful response from the authors. As I was thinking in my initial review, favoring faster clients may be of use in the iid settings. In the non-iid settings, it is important to not favor faster clients as they may lead to an optimum which is sub-optimal for the slower devices. Nevertheless, the non-iid setting is more realistic and your idea is important in combating this phenomena.
> >
> > The authors answered my questions for the s/n term as well as the performance of FAVAS in the non-iid setting. My only last remark is that the variance of FedAvg can grow due to the number of local SGD steps (I was unsure how many local steps you used). However, there still is synchronization with FedAvg after a certain number of steps, and so shouldn't the variance once again return to 0? This is still unclear to me.
> >
> > Overall, I thank the authors for their in depth responses which have addressed the majority of my questions. My score will remain the same.

---

### Official Review · Reviewer_yPyi · 2023-07-06

**Soundness:** 3 good
**Presentation:** 3 good
**Contribution:** 3 good
**Rating:** 6
**Confidence:** 4

**Summary:**

In this work, the authors propose an asynchronous communication protocol based on federated averaging, called FAVAS. The proposed method builds on top of a previously released method in asynchronous settings by addressing the bias and variance limitations of the previous approach through an unbiased estimator based on local reweighting. The authors make their claims clear with both theoretical and experimental results.

**Strengths:**

(1) The paper is easy to follow, with good structure and motivation.

(2) The paper also provides a thorough related work discussion, convergence analysis, and empirical evaluation.

(3) Theoretical and empirical variance analysis reduction for asynchronous settings through unbiased local model reweighing.


**Weaknesses:**

(1) Missing background makes it hard to understand essential concepts of the proposed framework.

(2) The work needs additional clarifications in the numerical evaluation and convergence bounds.

**Questions:**

Table 1 introduces the bounds of different methods. However, an extensive discussion of what each bound represents for each method is absent. There are a lot of notations and little semantic interpretation. This needs to be updated. What do the C_ constants represent for each method? Why is this different across methods when it only depends on client speeds, which are environment-specific?

Please clarify the reweighting factor $\alpha$; why the expectation is considered is unclear. Do you create a distribution based on the previously completed steps? What is the intuition behind the reweighting? Similarly, in the preliminaries section, what does the $\tilde{h}^i_{t,q}$ represent? It is hard to follow. Moreover, the motivation for introducing quantization in the proposed framework is not clearly conveyed. It is not evident why quantization is critical; some theoretical numerical results may enhance the contribution of this claim.

Can you elaborate on what the non-uniform timing experiments refer to? More details are needed for this concept. You introduce the global step (for the first time) in line 293. Is this step similar to an aggregator's update? In line 290, you set K=20 and state that K represents local epochs, while in your analysis, you considered K to be the local step. Are the two Ks the same? For the 10 conducted experiments in Table 2 did you set a different seed every time? If yes, please state that. Also, Table 2 shows that while FedBuff and FAVAS lose some performance as we go from more to less fast clients (2/3 fast vs. 1/9 fast), FedAvg and QuAFL seem to improve. Some discussion on this would be interesting. It would also be interesting to conduct some experiments where all clients.


When comparing the different methods in Figure 2, there is a high variance for the other baselines (FedBuff, QuAFL) at the beginning. Please provide some discussion in the main text regarding this and relate it to the units of time analysis of Table 1. It would be good also to point the reader to your corresponding Figure 6 in the appendix, presenting the variance magnitude for CIFAR-10.

An analysis of how different K values affect the computation and communication tradeoff is critical. From the current analysis, it is unclear what is a good choice for K.

Consider extending your related work to account for the FedRec asynchronous approach discussed in work[1], which weights local models based on steps staleness each client has performed and for the semi-synchronous protocol where the training workload is assigned based on the time it takes the slowest client to perform an epoch.

Comments:

line 108: Fedbuff -> FedBuff

line 139: term are -> is taken

line 137: we identified -> this could potentially reveal authorship

line 24 (Algorithm): equation (3) is never defined


[1] Semi-Synchronous Federated Learning for Energy-Efficient Training and Accelerated Convergence in Cross-Silo Settings. Dimitris Stripelis, Paul M. Thompson, José Luis Ambite.


**Limitations:**

Yes.

---

> ### Author Rebuttal · Authors · 2023-08-04
>
> We thank the reviewer for his time, careful reading, and insightful comments. We also thank the reviewer for highlighting that our paper "is easy to follow, with good structure and motivation".
>
> Q.1 "Table 1 " :   We agree Table.1 is dense and requires more details. First, each constant $C_$ depends on the client speed and represents the unit time to wait between two consecutive server steps. We agree that it is difficult to find a closed form for $C_$ (for FedAvg, this depends on the distribution of the slowest client speed; for AsyncSGD, this depends in a very complex way of the distribution of client speeds, which is captured in a very coarse way in the bounds by the "maximum" value of the delay, which is not a well-defined quantity - because under realistic assumptions of speed distribution, the delay has no reason to be bounded). But as explained earlier (l.243-248), we believe that complexity comparisons based only on the number of server steps are meaningless: what really counts is the physical time, as the time interval between two iterations at the central server can vary considerably between the different methods - if we express complexity as a function of the number of iterations at the central server, the FedAvg algorithm is optimal anyway, as we simply forget that it's the delay in updating the slowest clients that creates the difficulty. It is challenging to compare our rate with that of QuAFL which makes an additional assumption that all clients compute at least $E>0$ local steps on average [here again, the bound is clearly affected by the slowest clients]. We do not make such assumption and FAVAS is provably less affected by slow clients.
>
> Q.2 "clarify the reweighting factor $\alpha$..." :  The intuition behind the reweighting is that the stochastic gradient $\widetilde{h}_t^i$ computed by some client $i$ is a made of more local steps for fast clients. Indeed it is likely that fast client will be able to compute a large number of steps when being contacted, while slow clients will compute a smaller number. Hence, aggregating client contributions regardless the number of local steps done (what QuAFL does) will favor fast clients. As a consequence we propose to normalize the contribution of each client. Each sum of computed gradients $\widetilde{h}_t^i$ is reweighted by some $\alpha_i$. We have proposed two versions for $\alpha_i$:
>
> either we rescale the sum of gradients by the exact number of locally computed epochs (the random variable $E^{i}_{t+1} \wedge K$),
>
> or we rescale it by the expected value of the number of local epochs $\mathbb{E}[E^{i}_{t+1} \wedge K]$.
>
> Q.3 "It is not evident why quantization is critical" : We agree weight/activation quantization is not critical. We want to emphasize quantization will only increase (see Rk.1 and Rk.5) the variance of the stochastic gradients. To enhance the contribution of this claim (see Rk.6) we have trained a neural network with 4 bits precision arithmetic in Appendix (see Fig.7), and show that the results are close to their full-precision counterpart.
>
> Q.4 "what the non-uniform timing experiments refer" : Our indication is quite misleading, but the explanation is very simple. Non-uniform timing means that we consider clients with different speeds. We will rephrase this part.
>
> Q.5 "You introduce the global step (for the first time) in" You are right, this sentence refers to an aggregator's update. We will keep naming consistency, and rephrase it as "server step" to avoid any mistakes.
>
> Q.6 "Are the two Ks the same?" : You are right, this refer to the same $K$, i.e. the number of maximum local epoch a client can do. We will rephrase this sentence.
>
> Q.7 "did you set a different seed every time" : Yes we did. We will precise it on line.315.
>
> Q.8 "high variance for the other baselines (FedBuff, QuAFL) at the beginning" : FedBuff and QuAFL suffer from variance in between clients and server model initially for separated reasons. As explained line.310, during the first server steps, mainly fast clients will contribute in FedBuff. This results in a high model variance that is getting mitigated in the following steps when models updates are getting smaller. For QuAFL, the important client drift (see line.139) introduced by the client update (line.137) explains the high model variance. It reduces after some server steps, as the model updates (and hence the client drifts) are getting smaller.
>
> Q.9 "a good choice for K" : For FAVAS, the value of $K$ has no impact on the bandwidth. Indeed the central server chooses to interrupt some clients every $x$ seconds, no matter the number of local epochs they have computed so far. However $K$ has a deep impact on the complexity bounds. Choosing $K$ is a trade-off between how fast one wants to forget about the initial conditions (the term $F:=(f(w_0) - f_*)$) and the variance of the gradients computed (the terms $\sigma^2$ and $G^2$). The second and third order terms ($\epsilon^{-\frac{3}{2}}$ and $\epsilon^{-1}$) suggest to choose a small value for $K$. However the first order term (in $\epsilon^{-2}$) is decreasing with $K$, and depends on client speeds. When $K$ is large enough, this first order term (discarding the constant terms) can be approximated by $\frac{1}{\mathbb{E}[E_t^i]}$.
>
> Q.10 "FedRec" : Thank you for the reference! Our related works section mainly focuses on synchronous VS asynchronous methods. We will add a description about semi-synchronous approaches.
>
> Typos: Thank you for the typos lines.24(Algo), 108, 139. We will fix it.
> Regarding your remark line.137, we want to rephrase it to makes it clear. We do not want to reveal authorship, but we can guarantee we are not the authors of QuAFL (Zakerinia et al.,2022). However, in line.137 we want to say that we have identified one major issue in the work from (Zakerinia et al.,2022), that is the way they propose to compute the client update.
>
> We hope that these explanations will address your concerns. Thank you again for your support.

---

> > ### Comment · Reviewer_yPyi · 2023-08-16
> >
> > I have carefully read the authors' responses and the rest of the reviewers' comments. I sincerely appreciate the effort of the authors to address my concerns regarding different algorithmic notations (i.e., K, $\alpha$) and their impact and trade-off in learning performance. Overall, the authors have addressed all my major concerns. My score remains the same.

---

### Official Review · Reviewer_1shw · 2023-07-06

**Soundness:** 2 fair
**Presentation:** 2 fair
**Contribution:** 2 fair
**Rating:** 4
**Confidence:** 4

**Summary:**

The paper proposes a novel algorithm called FAVAS for asynchronous federated learning with heterogeneous clients, that is the modification of the previous QuaFL algorithm. The paper provides theoretical convergence rate of the proposed algorithm, and experimental evaluations of the effectiveness of their method.


**Strengths:**

- The studied question is important and well posed.

- Experimental results show a large improvement of FAVAS over the previous QuaFL, and some improvement over FedBuff.


**Weaknesses:**

Overall the paper is hard to read. The major concerns are unclear presentation of the theoretical results, and limited experimental evaluation.

1. In experiments the learning rate was set to some specific parameter 0.5, which is not a fair comparison. For a fair comparison, it is good to tune the learning rate separately for each of the methods. Same as for fair comparison it is good to tune the value of Z too. FedBuff proposed to set Z=10 when the delays are distributed close to the realistic FL system. However in your simulations the delays are distributed differently and thus the other values of Z might be optimal.

2. The convergence rates in Table 1 are very hard to compare between each other. First, there are no estimates of how large $C_{FedAvg}, C_{FedBuff}, C_{AsyncSGD}$. Second, it is unclear how large $\sum_{i = 1}^n a_i / n $ might be, so it is unclear if the FAVAS leading term of convergence is better or worse than previous rates.

3. The leading term  of convergence of FAVAS seems to be worse than QuAFL, as in QuaFL it is divided by E^2, while in FAVAS it is not divided by that.

4. It is also hard to understand how tight the proposed analysis is.

5. There are many other algorithms for asynchronous FL missing: such as R1, R2, R3. Experimental comparison to these baselines is also missing.

[R1] Gu et. al., Fast Federated Learning in the Presence of Arbitrary Device Unavailability,
[R2] Yan et al, Distributed Non-Convex Optimization with Sublinear Speedup under Intermittent Client Availability.
[R3] Glasgow et al, Asynchronous Distributed Optimization with Stochastic Delays.



**Questions:**

1. I did not understand the sentence in lines 86-87: “However, when staleness is …. “
2. Your algorithm is very similar to FedBuff with concurrency = n. I think your algorithm could also benefit from having concurrency smaller than n, especially when the number of nodes n is very large.


Minor:
1. On line 23 of Algorithm 1, the reference should be to the equation (1) instead of (3).
2. In Lemma 2, and line 207 there is a typo in the symbol above $h$.
3. Line 197: w_t^i is not defined.


**Limitations:**

-

---

> ### Author Rebuttal · Authors · 2023-08-04
>
> First, we thank the reviewer for his time, careful reading, and comments. We are pleased that the reviewer appreciates that the question "is well posed". We also thank the reviewer for highlighting our "Experimental results show a large improvement over the previous QuaFL".
> We answer to each comment following.
>
> Weaknesses.1 "it is good to set the learning rate separately" We agree that setting a hyperparameter to a fixed value does not allow for a fair comparison. For our experimental part, we reproduce the performance of the method we compare with, and for all algorithms, we have optimized the stepsize by grid search. We restarted the simulation for FedBuff on the MNIST dataset with a non-iid split (as described in our setting) and report the performance in the following table:
> |Z|learning rate|Accuracy|
> | -| -| -|
> |1|0.2|22.8|
> |1|0.7|15.2|
> |5|0.2|84.7|
> |5|0.7|43.4|
> |15|0.2|86.8|
> |15|0.7|29.1|
> We will add more details in the experimental section about our choice for hyperparameters.
>
> Weaknesses.[2,3,4] "The convergence rates in Table 1 are very difficult..." First, each constant $C_$ depends on the client speed and represents the unit time to wait between two consecutive server steps. We agree that it is difficult to find a closed form for the above constant terms (for FedAvg, this depends on the distribution of the slowest client speed; for AsyncSGD, this depends in a very complex way of the distribution of client speeds, which is captured in a very coarse way in the bounds by the "maximum" value of the delay, which is not a well-defined quantity - because under realistic assumptions of speed distribution, the delay has no good reason to be bounded). But as explained earlier (l.243-248), we believe that complexity comparisons based only on the number of server steps are meaningless: what really counts is the physical time, as the time interval between two iterations at the central server can vary considerably between the different methods - if we express complexity as a function of the number of iterations at the central server, the FedAvg algorithm is optimal anyway, as we simply forget that it's the delay in updating the slowest clients that creates the difficulty. It is challenging to compare our rate with that of QuAFL which makes an additional assumption that all clients compute at least $E > 0$ local steps on average [here again, the bound is clearly affected by the slowest clients]. We do not make such  assumption and FAVAS is provably less affected by slow clients.
>
> FAVAS remains attractive even in the homogeneous case where we assume that  (1) each client has the same average speed and therefore perform the same average number of local steps $E$ and (2)  $G=0$ (homogeneous loss functions). Under these assumptions, taking $K$ large enough, we get  $\sum\nolimits_{i=1}^n\frac{a_i}{n}$ is about $\frac{1}{E}$ ($+ o(\frac{1}{K^2}$)). Hence, $\sum\nolimits_{i=1}^n\frac{a_i}{n} \ll \frac{K}{E^2}$. As a consequence, the leading term (in $\epsilon^{-2}$) from FAVAS ($FL\sigma^2\sum\nolimits_{i=1}^n\frac{a_i}{n}$) is smaller than the leading term from QuAFL ($FL\sigma^2\frac{K}{E^2}$). We want to highlight the differences between FAVAS and QuAFL: first, as the local update strategy we propose (line.26 in Algo.1) is different from QuAFL's, the "contraction Lemma" (line.203) is different from QuAFL's, and our rate is better. Second, we want to emphasize that we do not assume clients do the same (on average) number of local steps, what QuAFL does, and we propose a new reweighting strategy with $\alpha_i$.
>
> Weaknesses.5 Thank you for the interesting references, we will add a discussion in the related works section. One can note all bounds in [R1, R2, R3] depend on some maximum delay $\tau_{max}^2$ (or equivalent quantities).  Under the assumptions we are making here, this quantity is a priori infinite - even if the processing time for each gradient step for a given client is bounded, as computation requests can accumulate and the number of requests is not almost surely bounded - : the complexity bounds reported are useless (see our discussion in line.234-242).
>
> Question.1 : Our formulation is misleading, we will reword it. We would like to emphasize that the server model in FedAsync (Algo.1, page.3, Xie et al., 2019) will not update if the staleness is too large (the function $s(t-\tau)$ is close to $0$ for large $t-\tau$). To mitigate this, the server must reduce the frequency of the scheduler that triggers the clients. This is the same bottleneck as FedAvg: a longer time must be waited between two consecutive server steps.
>
> Question.2 : The core idea of FAVAS (even with concurrency = n) is different from FedBuff. For each server step in FAVAS, we can choose the time to wait, say $\mathcal{T}$ seconds, and we interrupt all clients every $\mathcal{T}$ seconds even if they have not finished to compute the total number of gradient steps. However, FedBuff does not explicitly control the time elapsed between two consecutive server steps: if the slowest client takes $S$ seconds to compute a gradient, each server step will take $S$ seconds. Note also that FedBuff analysis depends on $\tau_{max}$ quantity, a quantity which is poorly defined (and unbounded; see additional pdf), and we would refer reviewers to a more detailed discussion in lines.232-242 of how it affects complexity bounds.
>
> Minor: 1) Thank you for the typo, we will fix it!
>
> 2) $\widetilde{h}$ is a biased estimator of the gradient. Our analysis (in Lemma 2, and line 207) deals with the *unbiased* gradient estimator defined as $ \check{h}_t^i$
>
> $ = \frac{1}{\alpha^i} \sum_{q=1}^{E_t^i \wedge K} \widetilde{h}_{t,q}^i $ in line 182.
>
> 3) We agree it is not clear that we are mentioning the local model of node $i$ at step $t$ (previously defined line 173). We will put an additional comment.
>
> We thank the reviewer again for his/her comments and support. We would be happy to provide more elements if there remain any unresolved questions.

---

> > ### Comment · Reviewer_1shw · 2023-08-16
> >
> > I would like to thank the authors for their detailed response to my questions. My remaining concerns are:
> >
> > 1. Thanks for the additional evaluation. However, the chosen set of hyperparameters is limited: maximum accuracy achieved for Z = 15, which is on the edge of the chosen range of Z, it seems that increasing Z further might further improve performance of FedBuff. As well as the learning rates were chosen only from two learning rates, making the optimal learning rate being on the edge of a tried set of parameters.
> >
> > 2. Could there be some estimation of which C_ constants are larger/smaller than the others and by how much? Maybe some experimental illustration ?

---

> > > ### Author Response · Authors · 2023-08-17
> > >
> > > We thank again the reviewer for his/her time and insightful comments.
> > >
> > > 1) In the additional evaluation, we chose hyperparameters close to the optimal ones. But in our initial experiments, we have tested a broader range of values for learning rate and $Z$. We did not report them because these extreme values did not lead good performances. For instance we took $Z=10$ and $lr=0.03$ and obtain an accuracy of $79.5$ % on MNIST dataset with a non-iid split (as described in our setting): we believe small learning rates let FedBuff central server model evolves too slowly. We also tested $Z=50$ (half of the total clients) and $lr=0.5$ and obtain an accuracy of $71.8$ %. In the latter case, the buffer is too large and the central server has to wait a lot between two consecutive steps. Hence, in the limited time ($5000$ in our experiments) we fixed, FedBuff is able to compute fewer steps when $Z=50$.
> > >
> > > 2) On average, we have $C_{FedAvg} > C_{FedBuff} > C_{AsyncSGD}$. Note that FedBuff does not allow clients to stack jobs (as opposed to AsyncSGD), but $C_{FedBuff, Z=1} = C_{AsyncSGD}$, and the value of $C_{FedBuff}$ increases with $Z$. These terms depend on the distribution of client speed. Under the setting we consider in our paper, we have: $C_{FedAvg} \simeq 898 \cdot C_{AsyncSGD}$, $C_{FedBuff, Z=5} \simeq 5.4 \cdot C_{AsyncSGD}$, $C_{FedBuff, Z=15} \simeq 16.3 \cdot C_{AsyncSGD}$.
> > >
> > >
> > > We hope that these explanations will address your concerns. Thank you again for your support.

---

### Author Rebuttal · Authors · 2023-08-04

We thank all reviewers for their insightful comments on our methodology and results. Below, we provide detailed responses to the questions raised by the reviewers and provide some additional comments on the scope and performance of our methodology.

We would like to emphasize the main contribution of our work: we propose a novel algorithm for training a FL system with different node speeds. We propose two unbiased versions for which server updates are not stalled by slow clients (unlike FedBuff). We also wish to emphasize that our performance bound does not depend on an intractable - and irrealistic - maximum delay (unlike FedBuff) and does not assume that clients have the same expected speed (unlike QuAFL). Please have a look at the additional pdf file that shows histogram of delays that were simulated for AsyncSGD/FedBuff, and highlights why the maximum delay assumption degrades the analysis. We also provide an experimental evaluation of the maximum delay that goes to $\infty$ when the number of server steps increases.

Last but not least, we understand that a fair comparison in Table 1 is not straightforward due to the $C_$ terms (expected value of the unit time to wait between two consecutive server steps). For FedAvg, this depends on the distribution of the slowest client speed; for AsyncSGD and FedBuff, this depends in a very complex way of the distribution of client speeds, which is captured in a very coarse way in the bounds by the "maximum" value of the delay, which is not a well-defined quantity - because under realistic assumptions of speed distribution, the delay has no good reason to be bounded. We believe that complexity comparisons based solely on the number of server steps obliterate a truly essential point: what really counts is the physical time, as the time interval between two iterations at the central server can vary considerably between the different methods (and FedAvg is optimal if we focus on the number of server steps). Our experimental results (Figure 5) show a large improvement over other approaches when accuracy is examined in terms of time rather than accuracy in terms of server steps.

---

### Comment · Area_Chair_nXuy · 2023-08-21

Dear authors,

Thank you very much for the rebuttal and discussion with the reviewers. I will incorporate all your comments during the subsequent discussion period before recommending a decision on the paper.

---

### Decision · Program_Chairs · 2023-09-21

**Decision:**

Reject

**Comment:**

This paper gives a convergence analysis of federated learning with asynchronous aggregation of local updates. There are several recent papers that seek to solve the same problem such as 'Sharper Convergence Guarantees on Federated Learning', 'FedBuff', and 'QuaFL', to name a few. The contribution of this paper is unclear in this somewhat crowded research area. While the reviewers appreciated the algorithm, analysis, and evaluation, they raised concerns about the writing of the paper and comparison with prior works. The paper is very much borderline with both positive and negative reviews, but based on the comments and discussion and my own reading of the paper, I recommend rejection. I hope that the authors can use the feedback to improve the paper.